physical chemistry/nanotechnology

absorption refrigeration, ionanofluids, thermal conductivity, heat transfer coefficient

**Author for correspondence:**
Fang-Fang Zhang
e-mail: zhangfangfang0629@126.com

This article has been edited by the Royal Society of Chemistry, including the commissioning, peer review process and editorial aspects up to the point of acceptance.

# Variations of thermophysical properties and heat transfer performance of nanoparticle-enhanced ionic liquids

Fang-Fang Zhang, Fei-Fei Zheng, Xue-Hong Wu, Ya-Ling Yin and Geng Chen

School of Energy and Power Engineering, Zhengzhou University of Light Industry, 136 Science Avenue, Gaoxin District, Zhengzhou 450001, People's Republic of China

(iD) F-FZ, 0000-0002-2710-5764

The ionic liquid (IL) 1-ethyl-3-methylimidazolium acetate ([EMIm]Ac) was investigated as a promising absorbent for absorption refrigeration. To improve the thermal conductivity of pure [EMIm]Ac, IL-based nanofluids (ionanofluids, INFs) were prepared by adding graphene nanoplatelets (GNPs). The thermal stability of the IL and INFs was analysed. The variations of the thermal conductivity, viscosity and specific heat capacity resulting from the addition of the GNPs were then measured over a wide range of temperatures and mass fractions. The measured data were fitted with appropriate equations and compared with the corresponding classical models. The results revealed that the IL and INFs were thermally stable over the measurement range. The thermal conductivity greatly increased with increasing mass fraction, while only slightly changed with increasing temperature. A maximum enhancement in thermal conductivity of 43.2% was observed at a temperature of 373.15 K for the INF with a mass fraction of 5%. The numerical results revealed that the dispersion of the GNPs in the pure IL effectively improved the local heat transfer coefficient by up to 28.6%.

## 1. Introduction

Environmental concerns and the global energy crisis have caused absorption refrigeration that has the advantages of reduced energy consumption, environmental friendliness and high efficiency to become a focus of international research [1,2]. In an absorption refrigerator, the absorber takes in the refrigerant from the evaporator and thereafter releases it to the condenser in the desorber accompanied by exothermic and endothermic effects. At present, the commonly used working pairs in absorption refrigeration cycles are aqueous solutions of lithium bromide

($H_2O$/LiBr) [3] and ammonia ($NH_3$/$H_2O$) [4]. Unfortunately, their broader industrial application has been hindered by the inherent defects of crystallization, corrosion, high working pressure and toxicity [5]. Therefore, the discovery and development of new working pairs is crucial. Over recent years, ionic liquids (ILs) have been widely studied as new environmentally friendly solvents for various applications [6–9], owing to their negligible volatility, high gas solubility and good thermal stability [10]. ILs have attracted remarkable attention in the field of absorption refrigeration since Shiflett & Yokozeki [10] first proposed the use of ILs as absorbents for absorbing the refrigerant. To date, research into ILs as the absorbents for absorption refrigeration has mainly focused on the study of their physico-chemical properties and thus, the selection of potential working pairs for industrial applications. Cao & Mu [11] reported that the cation dependence of the water absorption ability of ILs can typically be ranked as follows: imidazolium > pyridinium > phosphonium; similarly, the water sorption capacity, rate and difficulty to reach equilibrium at 23°C and 52% (relative humidity) for the investigated ILs with [BMIM] cation, was approximately as follows: [Ac] > [Cl] > [Br] > [TFA] (trifluoroacetic acid) > [$NO_3$] > [TFO] (trifluoromethanesulfonate) > [$BF_4$] > [$Tf_2N$] (*bis*((trifluoromethyl)sulfonyl)imide) > [CHO] > [$PF_6$]. Su *et al.* [12] studied the absorption refrigeration cycle using a new working pair consisting of an IL and water. The results indicated that, compared with the typical working pair of $H_2O$/LiBr, the single-stage absorption cycle using aqueous 1-ethyl-3-methylimidazolium acetate ([EMIm]Ac) exhibited almost the same coefficient of performance at a generation temperature of 100°C and a slightly higher performance at higher temperatures. These results demonstrated the feasibility of applying the working pair of [EMIm]Ac/$H_2O$ to absorption refrigeration. Current research on [EMIm]Ac is confined to theoretical refrigeration performance analysis based on the enthalpy–humidity diagram [12]. However, during a practical absorption refrigeration cycle, the processes of absorption by and desorption from the absorbent are often performed under cooling and heating, respectively. The cooling and heating efficiency directly affects the absorption and desorption efficiency. In particular, good heat transfer performance is required for the [EMIm]Ac on account of the performance feature of [EMIm]Ac gained from [12]. Whereas, He *et al.* [13] measured the thermal conductivity of the IL [HMIM]$BF_4$ in the range of 303–453 K and reported that it ranged from 0.167 to 0.197 W $m^{-1}$ $K^{-1}$. Oster *et al.* [14] measured the thermal conductivities of five ILs, such as butanoate [$P_{14,6,6,6}$][ButO], hexanoate [$P_{14,6,6,6}$][HexO] and decanoate [$P_{14,6,6,6}$][DecO], in the temperature range of 283–373 K. The thermal conductivities of this series of ILs were found to be within the range of 0.147–0.162 W $m^{-1}$ $K^{-1}$. It is apparent from these reports that the thermal conductivities of ILs are generally low. Therefore, it would be of great value to enhance the thermal conductivity of ILs such as [EMIm]Ac to allow their use as absorbents. In recent years, researchers have been able to increase the thermal conductivity of ILs by adding nanophases to form IL-based nanofluids (ionanofluids, INFs) [15,16]. Commonly used nanophases have included silica, nanosized carbons, metals, metal oxides, nitrides, carbides and graphene [17,18]. Among these, graphene is a novel carbon-based nanomaterial with excellent thermal, electronic and mechanical properties. The thermal conductivity of graphene is as high as approximately 5000 W $m^{-1}$ $K^{-1}$, which makes it a very promising nanoadditive for nanofluids [19] (table 1).

In the present study, the use of the promising absorbent [EMIm]Ac in an absorption refrigeration cycle was evaluated. The thermal stability, viscosity, thermal conductivity and specific heat capacity were measured. On account of the low thermal conductivity of [EMIm]Ac, graphene nanoplatelets (GNPs) were dispersed in the IL to obtain INFs. The variation of the thermal conductivity, viscosity and specific heat capacity of the INFs was analysed. The measured data were fitted with equations and compared with the corresponding classical models. Finally, considering that the most frequently used flow mode in the absorber and desorber units is falling film flow [20,21], the variation of the falling film heat transfer coefficient of the nanoparticle-enhanced IL in a horizontal tube was numerically evaluated.

# 2. Material and methods

## 2.1. Chemicals and materials

### 2.1.1. Ionic liquid

[EMIm]Ac (purity ≥ 98%, water content ≤ 1%) was purchased from the Lanzhou Institute of Chemical Physics, Chinese Academy of Sciences. The structural formula of [EMIm]Ac is depicted in figure 1 and its thermophysical properties measured in this work at the standard temperature of 293.15 K are summarized in table 2. The minor deviations in certain physical properties were ascribed to the different manufacturing processes used by different suppliers.

**Figure 1.** Structural formula of [EMIm]Ac.

**Table 1.** Nomenclature.

| | |
|---|---|
| $C_P$ | specific heat capacity, J kg$^{-1}$ K$^{-1}$ |
| $D$ | tube diameter, m |
| **F** | body force, N |
| $h$ | heat transfer coefficient, W m$^{-2}$ K$^{-1}$ |
| **g** | gravity acceleration, m s$^{-2}$ |
| $q$ | heat flux density, W m$^{-2}$ |
| $T$ | temperature, K |
| **u** | velocity vector, m s$^{-1}$ |
| **Greek** | |
| $\lambda$ | thermal conductivity, W m$^{-1}$ K$^{-1}$ |
| $\mu$ | dynamic viscosity, kg m$^{-1}$ s$^{-1}$ |
| $\rho$ | density, kg m$^{-3}$ |
| $\varphi$ | volume fraction |
| $\Gamma$ | liquid film flow rate on one side of the tube per unit length, kg m$^{-1}$ s$^{-1}$ |

**Table 2.** Physical properties of [EMIm]Ac at 293.15 K.

| property | present study | literature | deviation (%) |
|---|---|---|---|
| molecular weight | 170.2 | – | – |
| density (g cm$^{-3}$) | 1.10493 | 1.10302 [22] | 0.17 |
| viscosity (mPa s) | 155.1 | 162 [23] | 4.26 |
| specific heat capacity (J kg$^{-1}$ K$^{-1}$) | 1868 | 1625 [12] | 14.5 |
| thermal conductivity (W m$^{-1}$ K$^{-1}$) | 0.221 | 0.211 [24] | 4.7 |

## 2.1.2. Graphene nanoplatelets

The GNPs were purchased from Chengdu Organic Chemicals Co. Ltd, Chinese Academy of Sciences. The GNPs exhibited a thermal conductivity of 3000 W m$^{-1}$ K$^{-1}$, a diameter of 5–10 µm, a thickness of 4–20 nm and a density of 0.6 g cm$^{-3}$ and consisted of less than 20 layers (the data come from the manufacturer). A thermal field emission scanning electron microscopy (JSM-7001F, JEOL, Japan) image of the GNPs is presented in figure 2. It can be seen that the GNPs possessed the expected sheet structure.

## 2.2. Synthesis of ionanofluids

A series of INFs was prepared using a two-step method. Various mass fractions of the GNPs were dispersed in [EMIm]Ac using a constant temperature magnetic stirrer for 60 min, followed by ultrasonication for 60 min at 25°C by ultrasonic cleaning machine (C15, XIERBAO, Beijing, power: 300 W, frequency: 40 kHz) for 60 min, affording INFs with mass fractions of 0.05, 0.1, 0.3, 0.5, 1, 2, 3, 4 and 5%.

## 2.3. Measurement methods

Thermal conductivities were evaluated using a laser thermal conductivity meter (LFA 467, NETZSCH, Germany) by the flash method over the temperature range of 293.15–373.15 K. Pyroceram 9606

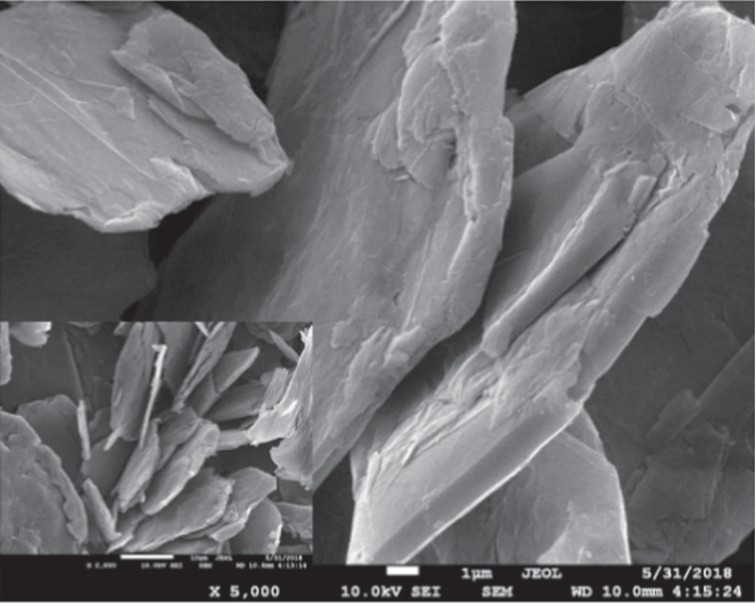

**Figure 2.** Thermal field emission scanning electron microscopy image of the GNPs.

provided by the supplier was used to calibrate the meter with a relative uncertainty of 3%. Viscosities were measured using a viscotester (Viscotester iQ, Haake, Germany) over the temperature range of 283.15–373.15 K. The viscotester was calibrated by the standard viscosity liquid provided by the supplier with a relative uncertainty of 0.65%. The torque resolution is 0.01 mN m. The type of geometry used is cylinder double-gap (inner cylinder: inner diameter and outer diameter are 20.810 and 21.281 mm; outer cylinder: inner diameter and outer diameter are 26.594 and 27.200 mm; height: 40 mm; distance: 4 mm; volume: 3 cm$^3$). The shear rate of 500 s$^{-1}$ was selected to avoid the Taylor vortices area for steady-state shear testing. Specific heat capacities were determined using a differential scanning calorimeter (DSC 214 Polyma, NETZSCH, Germany) based on the sapphire method over the temperature range of 303.15–383.15 K. This was calibrated using sapphire provided by the supplier with a relative uncertainty of 0.5%. Thermal stabilities were analysed using the same differential scanning calorimeter as above, and the samples were heated from −50 to 350°C at a rate of 10°C min$^{-1}$ under a nitrogen atmosphere. Densities were measured using a densimeter (DMA 5000M–Lovis 200M, Anton Paar Co., Austria). This was calibrated using air and ultrapure water provided by Anton Paar GmbH and compared with values reported in the densimeter instruction manual. The found uncertainty was less than $\pm 1 \times 10^{-5}$ g cm$^{-3}$ and the accuracy is $5 \times 10^{-6}$ g cm$^{-3}$.

## 2.4. Numerical approach and reliability validation

Considering that the most frequently used flow mode in the absorber and desorber units is falling film flow on a horizontal tube and its symmetrical structure, the physical model of falling film flow on half of the horizontal tube based on Gambit is depicted in figure 3. The boundary conditions are labelled. A slot with width of 3.0 mm was used as the liquid distributor and this was set at the left-most part of the solution domain. The distributor inlet was set as the velocity inlet. The liquid and gas phases were water and air, respectively. The simulated water entered from the distributor hole and then flowed around the smooth tube in the air atmosphere at a temperature of 298 K and a pressure of 101.325 kPa. The solution domain was discretized by quadrilateral elements. The areas near the tube and liquid inlet were refined. The volume of fluid model was selected for the simulations, which were performed using the Fluent software (v. 6.3.26). The governing equations can be expressed as follows [25]:

$$\nabla(\mathbf{u}) = 0, \tag{2.1}$$

$$\frac{\partial(\rho \mathbf{u})}{\partial t} + \nabla(\rho \mathbf{u} \cdot \mathbf{u}) = \nabla(\mu \nabla \mathbf{u}) - \nabla(p) + \rho \mathbf{g} + \mathbf{F} \tag{2.2}$$

and

$$\frac{\partial(\rho T)}{\partial t} + \nabla(\rho \mathbf{u} T) = \nabla \left( \frac{\lambda}{c_P} \nabla T \right). \tag{2.3}$$

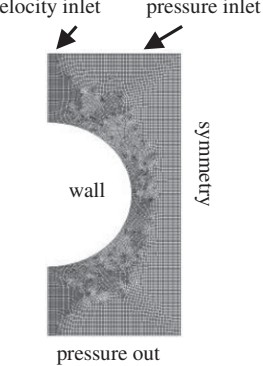

**Figure 3.** Physical model of falling film flow on a horizontal tube.

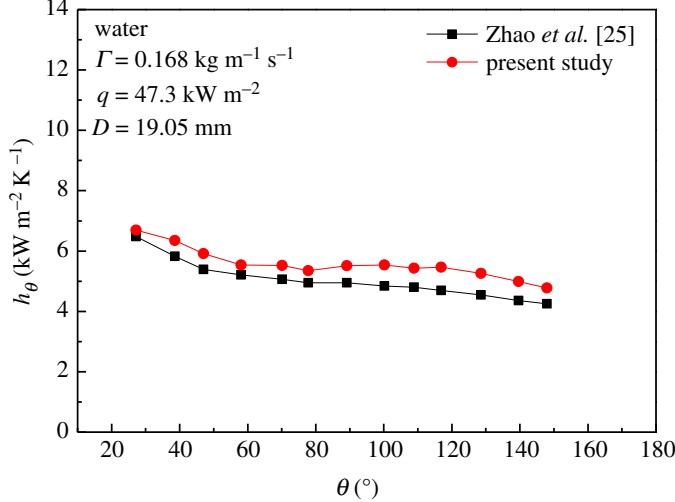

**Figure 4.** Comparison of the local heat transfer coefficient of water between the present and reference results.

where $\mathbf{u}$ is the velocity vector, $\rho\mathbf{g}$ is the gravitational force, $\mathbf{F}$ is the external body force of surface tension and $T$ is temperature.

Figure 4 shows a comparison of the local heat transfer coefficient of water between the present results and the reference results [25]. The tube diameter $D$ and distribution height were 19.05 mm and 6.3 mm, respectively. The heat flux density $q$ was 47.3 kW m$^{-2}$ and the liquid film flow rate on one side of the tube per unit length $\Gamma$ was 0.168 kg m$^{-1}$ s$^{-1}$. From the curves, it can be seen that the obtained numerical data were in good agreement with the reference data.

# 3. Results and discussion

## 3.1. Thermal stability of ionanofluids

Figure 5 presents the DSC curves for the base ionic liquid (BL), 0.05% INF and 2% INF at temperatures ranging from about −50 to 350°C, in which the initial endothermic points were analysed at 216°C, 197.17°C and 226.24°C, respectively. The decomposition temperatures of the BL, 0.05% INF and 2% INF were approximately 225.87°C, 226.72°C and 226.24°C, respectively. Hence, it can be concluded that the IL and INFs were thermally stable over the studied temperature range.

## 3.2. Viscosity of ionanofluids

The shear stress is plotted in figure 6a as a function of shear rate for the samples within the shear rate range of 1–500 s$^{-1}$ at 293.15 K, from which it can be found that the behaviour of INFs was quite Newtonian when the mass fraction of GNPs was lower than 0.5%. However, the behaviour of INFs

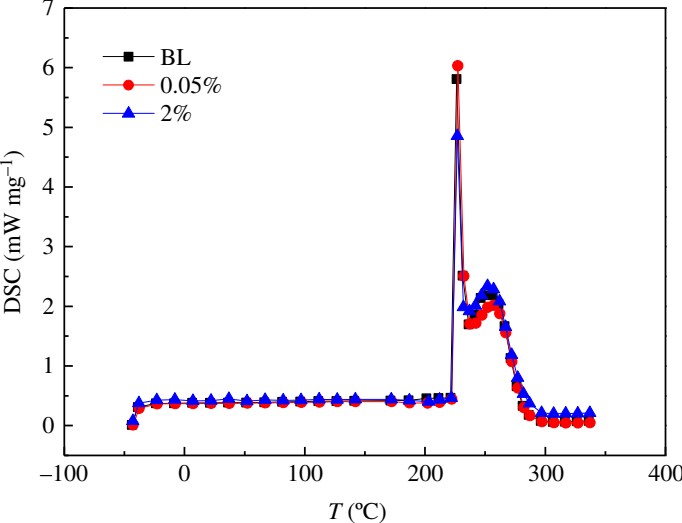

**Figure 5.** DSC curves for the BL, 0.05% INF and 2% INF at temperatures ranging from −50 to 350°C.

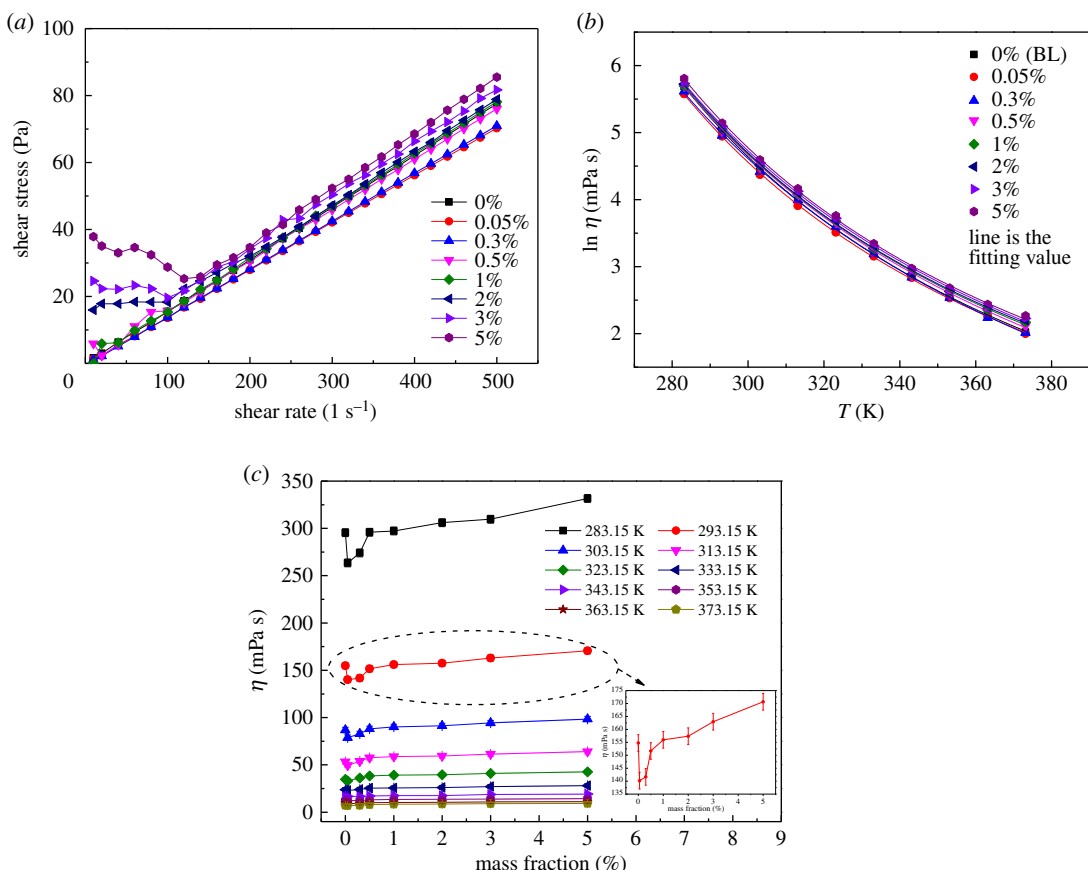

**Figure 6.** (a) The flow curves of the viscosity for the INFs, (b) natural logarithm of viscosity as a function of temperature and (c) viscosity versus mass fraction.

was quite non-Newtonian when the mass fraction of GNPs was larger than 0.5%. The variation of the natural logarithm of the viscosity (tested with a shear rate of $500 \, s^{-1}$) of the BL and the INFs with mass fractions of 0.05, 0.3, 0.5, 1, 2, 3 and 5% as a function of temperature is shown in figure 6b. It can be seen that in the studied temperature range, the viscosities of the INFs sharply decreased with increasing temperature. Figure 6c shows the viscosity as a function of the mass fraction. It can be seen that the viscosities of the INFs were lower than that of the base solution when the mass fraction was

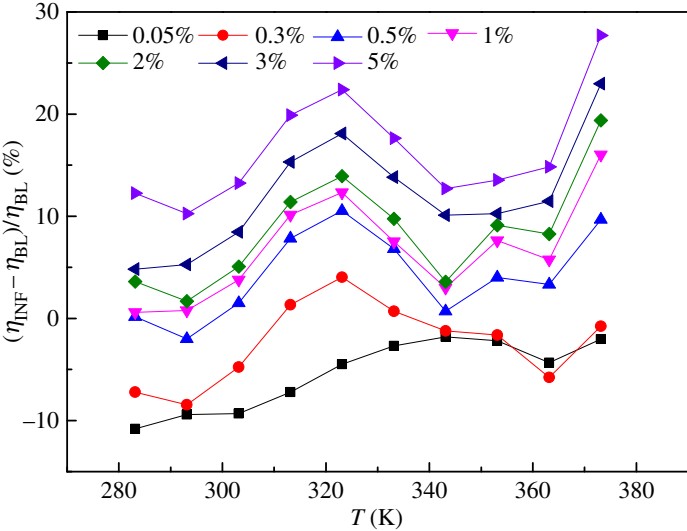

**Figure 7.** Deviation of the viscosities of the INFs from that of the BL.

**Table 3.** Fitting parameters of the VFT equation for the obtained viscosity data.

| mass fraction (%) | parameter | | | $R^2$ |
| --- | --- | --- | --- | --- |
| | $A_0$ | $A_1$ | $A_2$ | |
| 0 (BL) | −2.82633 | 1020.85237 | 163.24164 | 0.99995 |
| 0.05 | −2.85375 | 1039.69227 | 159.70455 | 0.99981 |
| 0.3 | −3.68272 | 1322.41148 | 140.56527 | 0.99956 |
| 0.5 | −3.05676 | 1127.57483 | 154.05057 | 0.99927 |
| 1 | −2.82684 | 1066.35243 | 158.00433 | 0.99919 |
| 2 | −2.59611 | 997.7974 | 163.16301 | 0.99899 |
| 3 | −2.79799 | 1075.99513 | 157.04136 | 0.99913 |
| 5 | −2.61161 | 1018.78783 | 162.04418 | 0.99897 |

less than 0.5%. This behaviour mainly originated from the dominant self-lubrication effect of GNPs at lower temperatures and mass fractions [26]. Owing to the higher viscosity of the IL, the slight change in viscosity was not obvious in the curves at mass fractions exceeding 1%. In addition, the effect of temperature on viscosity became weaker with increasing temperature. A maximum increase in viscosity of approximately 27.7% was observed for the INFs compared with the BL at a temperature of 373.15 K and a mass fraction of 5%, within the scope of this experiment. The relationship between the natural logarithm of the viscosity and the temperature was fitted using the Vogel–Fulcher–Tammann (VFT) equation (equation (3.1)). Table 3 summarizes the values of the fitting parameters $A_0$, $A_1$ and $A_2$ for the various mass fractions.

$$\ln \eta = A_0 + \frac{A_1}{T - A_2}. \tag{3.1}$$

In addition, the deviation of the viscosities of the INFs from that of the BL $((\eta_{INF} - \eta_{BL})/\eta_{BL})$ was analysed, as shown in figure 7. The deviation was found to fluctuate within approximately 20%, depending on the temperature and mass fraction. It can be concluded that both the mass fraction and the temperature had little effect on the viscosity deviation.

Furthermore, the measured values of viscosity were compared with the classical models of Einstein, Brinkman and Batchelor [27] and a suggested correlation was developed by modifying the factor in the original Einstein model for spheres from a value of 2.5 to a fitted value of 1.1, as shown in figure 8, which can well represent the GNPs in our study except at a lower mass fraction (with a maximum deviation of

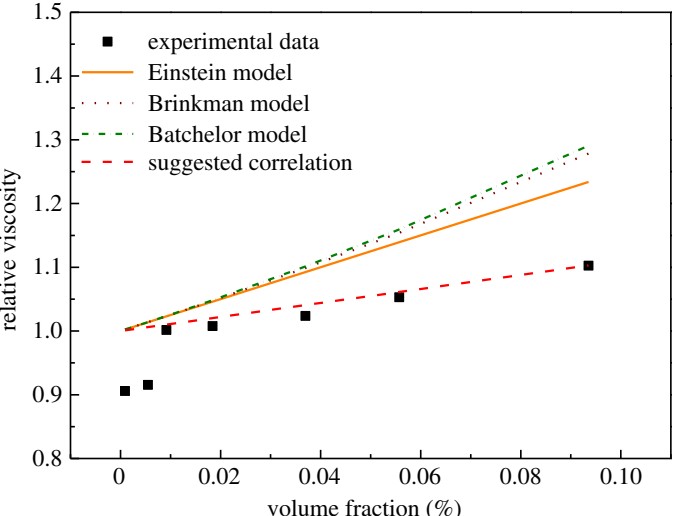

**Figure 8.** Comparison between the measured values of viscosity for different volume fraction of INFs at 303.15 K with the classical models for viscosity of Einstein, Brinkman and Batchelor and with the suggested correlation.

9.5%) due to a dominant self-lubrication effect of GNPs at lower mass fractions [26].

$$\text{Einstein model} \quad \frac{\eta_{INF}}{\eta_{BL}} = 1 + 2.5\varphi, \tag{3.2}$$

$$\text{Brinkman model} \quad \frac{\eta_{INF}}{\eta_{BL}} = \frac{1}{(1-\varphi)^{2.5}}, \tag{3.3}$$

$$\text{Batchelor model} \quad \frac{\eta_{INF}}{\eta_{BL}} = 6.5\varphi^2 + 2.5\varphi + 1, \tag{3.4}$$

$$\text{suggested correlation} \quad \frac{\eta_{INF}}{\eta_{BL}} = 1 + 1.1\varphi \tag{3.5}$$

and

$$\varphi = \frac{\omega\rho_{INF}}{\rho_{NP}}, \tag{3.6}$$

where $\eta_{INF}$ is the viscosity of the INF, $\eta_{BL}$ is the viscosity of the BL, $\varphi$ is the particle volume fraction calculated using equation (3.6), $\omega$ is the BL, $\varphi$ is the volume fraction of INF, and $\rho_{NP}$ and $\rho_{INF}$ are the densities of the nanoparticle and the INF, respectively.

Overall, the viscosity of the IL was higher. The addition of a small amount of the GNPs did not increase the viscosity. Conversely, the addition of the GNPs slightly decreased the viscosity of the INFs. It is also worth noting that heating dramatically reduced the viscosity of the INFs.

## 3.3. Thermal conductivity of ionanofluids

The thermal conductivities of the BL and the INFs with mass fractions of 0.05, 0.1, 0.3, 0.5, 1, 2, 3, 4 and 5% as a function of temperature are shown in figure 9a. It can be seen that in the studied temperature range, the temperature exerted little influence on the thermal conductivity of the INFs. The thermal conductivity increased significantly with increasing mass fraction. This phenomenon can also be observed in figure 9b. These results demonstrate that the addition of the GNPs significantly increased the thermal conductivity of the IL. However, it is worth noting that the addition of excess GNPs was not conducive to increase the thermal conductivity, as the dispersion of the GNPs in the IL gradually decreased with increasing mass fraction. In addition, a linear equation (equation (3.7)) was used to fit the experimentally measured thermal conductivity data. Table 4 summarizes the values of the fitting parameters $B_0$ and $B_1$ for the various mass fractions.

$$\lambda = B_0 + B_1 T. \tag{3.7}$$

The deviation of the thermal conductivities of the INFs from that of the BL $((K_{INF} - K_{BL})/K_{BL})$ was also analysed, as shown in figure 10. It is apparent that the temperature had little effect on the thermal conductivity. The maximum increase observed in the thermal conductivity of the INFs was

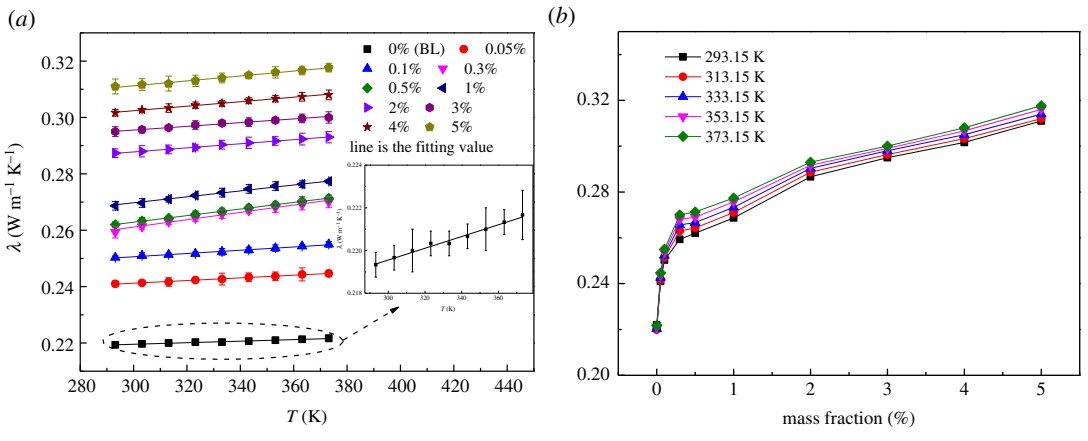

**Figure 9.** Thermal conductivity of the INFs: (*a*) thermal conductivity versus temperature and (*b*) thermal conductivity versus mass fraction.

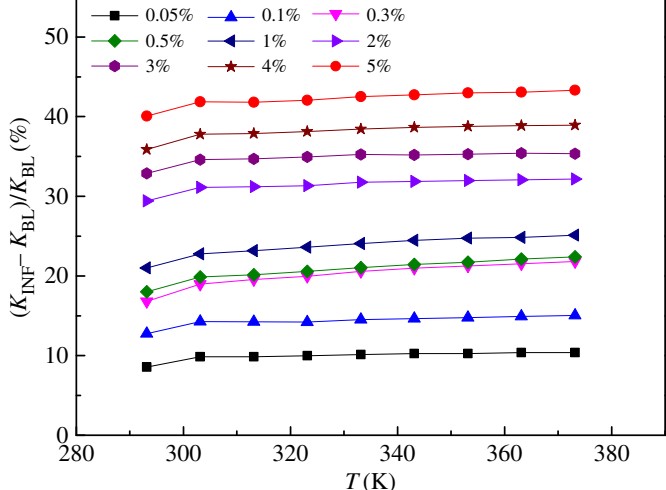

**Figure 10.** Deviation of the thermal conductivities of the INFs from that of the BL.

**Table 4.** Fitting parameters for the thermal conductivity data.

| mass fraction (%) | parameter | | |
| --- | --- | --- | --- |
| | $B_0$ | $B_1$ | $R^2$ |
| 0 (BL) | 0.2113 | $2.75146 \times 10^{-5}$ | 0.9814 |
| 0.05 | 0.22705 | $4.72167 \times 10^{-5}$ | 0.99584 |
| 0.1 | 0.23324 | $5.78453 \times 10^{-5}$ | 0.98808 |
| 0.3 | 0.222 | $1.30405 \times 10^{-4}$ | 0.98572 |
| 0.5 | 0.22774 | $1.17001 \times 10^{-4}$ | 0.99912 |
| 1 | 0.23819 | $1.05179 \times 10^{-4}$ | 0.99457 |
| 2 | 0.266 | $7.25622 \times 10^{-5}$ | 0.99677 |
| 3 | 0.27571 | $6.60535 \times 10^{-5}$ | 0.9871 |
| 4 | 0.27836 | $8.00711 \times 10^{-5}$ | 0.99661 |
| 5 | 0.28515 | $8.69292 \times 10^{-5}$ | 0.99291 |

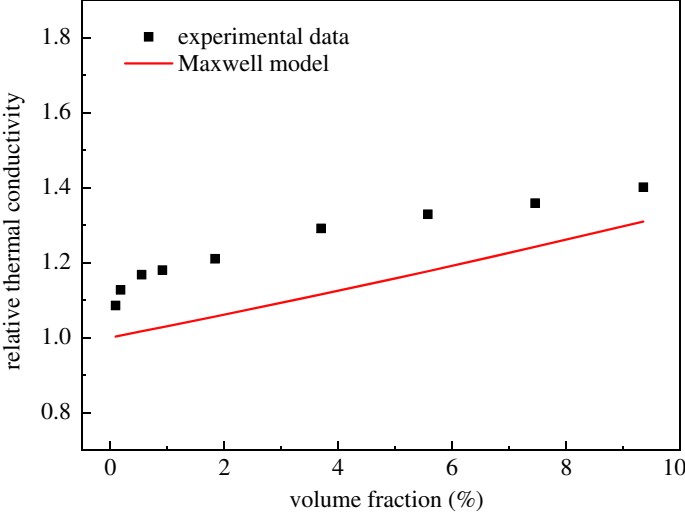

**Figure 11.** Comparison of the predicted relative thermal conductivity with the experimental data for the INFs at 293.15 K.

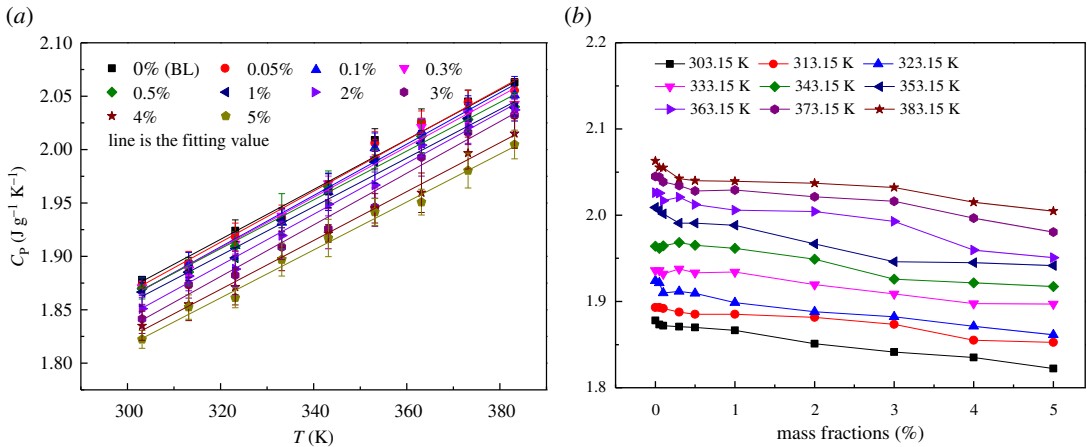

**Figure 12.** Specific heat capacity of the INFs: (*a*) specific heat capacity as a function of temperature and (*b*) specific heat capacity versus mass fraction.

43.2%, which was obtained for a mass fraction of 5% and a temperature of 373.15 K, within the scope of the experiment.

Furthermore, the relative thermal conductivity ($K_{\mathrm{INF}}/K_{\mathrm{BL}}$) at 293.15 K was compared with the Maxwell model [28] (equation (3.8)), as shown in figure 11.

$$\frac{K_{\mathrm{INF}}}{K_{\mathrm{BL}}} = \frac{K_{\mathrm{NP}} + 2K_{\mathrm{BL}} - 2\varphi(K_{\mathrm{BL}} - K_{\mathrm{NP}})}{K_{\mathrm{NP}} + 2K_{\mathrm{BL}} + \varphi(K_{\mathrm{BL}} - K_{\mathrm{NP}})},$$

(3.8)

where $K_{\mathrm{INF}}$ is the thermal conductivity of the INF, $K_{\mathrm{BL}}$ is the thermal conductivity of the BL, $K_{\mathrm{NP}}$ is the thermal conductivity of the nanoplatelets and $\varphi$ is the particle volume fraction calculated according to equation (3.6). The maximum error between the measurement values and the predicted data from the Maxwell model was 15.7%.

## 3.4. Specific heat capacity of ionanofluids

The specific heat capacities of the BL and the INFs with mass fractions of 0.05, 0.1, 0.3, 0.5, 1, 2, 3, 4 and 5% as a function of temperature are shown in figure 12*a*. The specific heat capacity of the INFs increased linearly with increasing temperature. Therefore, a linear equation (equation (3.9)) was used to fit the experimentally measured specific heat capacity data. Table 5 summarizes the values of the fitting parameters $C_0$ and $C_1$ for the various mass fractions. The influence of the mass fraction on the specific heat capacity is shown in figure 12*b*. The specific heat capacity of the INFs decreased with increasing

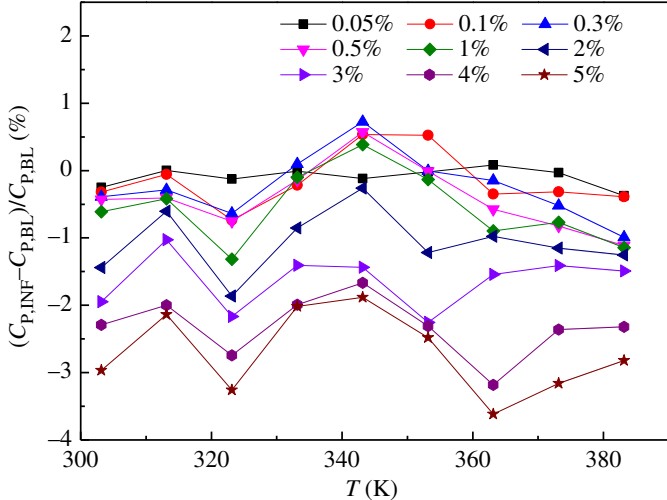

**Figure 13.** Deviation of the specific heat capacities of the INFs from that of the BL.

**Table 5.** Fitting parameters for the specific heat capacity data.

| mass fraction (%) | parameter | | $R^2$ |
| --- | --- | --- | --- |
| | $C_0$ | $C_1$ | |
| 0 (BL) | 1.16894 | 0.00233 | 0.99643 |
| 0.05 | 1.14696 | 0.00239 | 0.9931 |
| 0.1 | 1.13925 | 0.00241 | 0.98938 |
| 0.3 | 1.15168 | 0.00236 | 0.98816 |
| 0.5 | 1.17613 | 0.00228 | 0.98946 |
| 1 | 1.17216 | 0.00228 | 0.99003 |
| 2 | 1.12815 | 0.00239 | 0.99105 |
| 3 | 1.10699 | 0.00242 | 0.98498 |
| 4 | 1.13711 | 0.00229 | 0.99536 |
| 5 | 1.14209 | 0.00225 | 0.99146 |

mass fraction owing to the low specific heat capacity of the GNPs. A maximum reduction of approximately 3.62% was observed for a temperature of 363.15 K and a mass fraction of 5%, within the scope of the experiment. In an absorption refrigeration cycle, a lower specific heat capacity is beneficial for the temperature variation of the absorbent in both the absorber and desorber units.

$$C_P = C_0 + C_1 T. \tag{3.9}$$

The deviation of the specific heat capacities of the INFs from that of the BL ($(C_{P,INF} - C_{P,BL})/C_{P,BL}$) was also analysed, as shown in figure 13. In general, the deviation of the specific heat capacity of the INFs increased with increasing mass fraction. However, no obvious trend was evident in the value of $(C_{P,INF} - C_{P,BL})/C_{P,BL}$ as the temperature was varied.

Furthermore, the relative specific heat capacity ($C_{P,INF}/C_{P,BL}$) at 303.15 K was compared with the existing theoretical model [29] (below equation), as shown in figure 14.

$$C_{P,INF} = \omega C_{P,NP} + (1 - \omega)C_{P,BL}, \tag{3.10}$$

where $C_{P,INF}$ is the specific heat capacity of the INF, $C_{P,NP}$ is the specific heat capacity of the nanoplatelets, $C_{P,BL}$ is the specific heat capacity of the BL and $\omega$ is the mass fraction of the nanoplatelets.

As can be seen from figure 14, the reliability of the model was very high, and the maximum error between the measurement value and the predicted data from the model was only 1.3%.

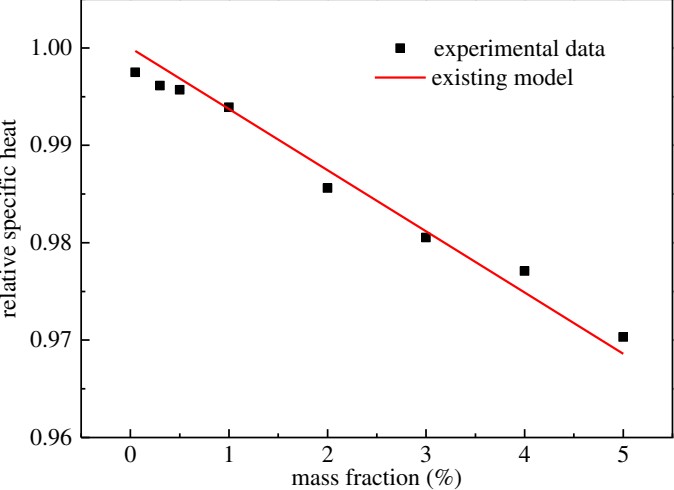

**Figure 14.** Comparison of the predicted relative specific heat capacity with the experimental data for the INFs at 303.15 K.

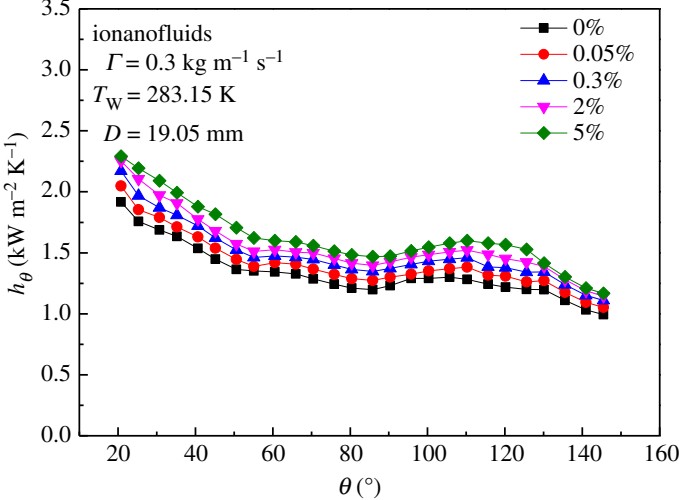

**Figure 15.** Local heat transfer coefficients of the falling films with different mass fractions of GNP.

**Table 6.** Properties of the working medium at 343.15 K measured in this work.

| property | [EMIm]Ac | 0.05% INF | 0.3% INF | 2% INF | 5% INF |
| --- | --- | --- | --- | --- | --- |
| density (kg m$^{-3}$) | 1073.6 | 1075.542 | 1076.404 | 1084.654 | 1095.003 |
| viscosity (Pa s) | 0.017311 | 0.017004 | 0.0171 | 0.01793 | 0.01951 |
| specific heat capacity (J kg$^{-1}$ K$^{-1}$) | 1964 | 1961.66 | 1968.2 | 1948.9 | 1917.21 |
| thermal conductivity (W m$^{-1}$ K$^{-1}$) | 0.22 | 0.243 | 0.268 | 0.291 | 0.315 |

## 3.5. Numerical results and discussion

Figure 15 shows a comparison of the time-averaged local heat transfer coefficients of the falling films using the pure IL and the INF with mass fractions of 0.05, 0.3, 2 and 5% as the absorbent. The properties of the working medium measured in this work are summarized in table 6. From the curves, it can be seen that the local heat transfer coefficient was highest for 5 wt% INF. The greatest increase in the heat transfer coefficient was 28.6%, with an increase in the thermal conductivity of 43.2%. It can be concluded that the addition of GNPs to the pure IL effectively improved the local heat transfer coefficient of the film falling along the horizontal tube.

# 4. Conclusion

INFs with mass fractions of 0.05, 0.1, 0.3, 0.5, 1, 2, 3, 4 and 5% were prepared. The thermal conductivities, viscosities and specific heat capacities of the INFs were measured and analysed at various temperatures. The results demonstrated that the addition of GNPs clearly increased the thermal conductivity of the INFs while decreasing the specific heat capacity and viscosity at lower mass fractions. The viscosity and specific heat capacity sharply decreased and increased, respectively, with increasing temperature, while the thermal conductivity only slightly changed. Within the scope of the experiment, the maximum increase in viscosity of approximately 27.7% for the INFs compared with the BL was achieved at a temperature of 373.15 K and a mass fraction of 5%; a suggested correlation was developed by modifying the factor in the original Einstein model for spheres from a value of 2.5 to a fitted value of 1.1 with the maximum deviation of 9.5%. The maximum increase in thermal conductivity of approximately 43.3% occurred at 373.15 K for the INF with a mass fraction of 5%; the error between the thermal conductivity measurement results and the predicted data from the Maxwell model was within 15.7%. The maximum reduction in specific heat capacity of approximately 3.62% was observed at a temperature of 363.15 K and a mass fraction of 5%; the error between the specific heat capacity measurement results and the predicted data from the existing model was within 1.3%. Finally, it was found that the local heat transfer coefficient increased by 28.6% compared with the pure IL when the INF with a mass fraction of 5% was used as the absorbent.

Data accessibility. Data available from the Dryad Digital Repository at: https://doi.org/10.5061/dryad.hv18jn6 [30].

Authors' contributions. F.-F.Zha. and F.-F.Zhe. designed and coordinated the study, analysed the data and wrote the manuscript; X.-H.W., Y.-L.Y. and G.C. collected the data. All authors gave final approval for publication.

Competing interests. We have no competing interests.

Funding. We are grateful for the financial support from the National Natural Science Foundation of China (grant no. 51606174), the Key scientific research projects in colleges and universities in Henan Province (grant no. 17A470017) and the Innovative Research Team (in Science and Technology) in University of Henan Province (grant no. 17IRTSTHN029).

Acknowledgements. We thank our anonymous reviewers for their insightful and helpful comments.

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
