## [Reviewer comments · Royal Society Open Science]

Review History

RSOS-182040.R0 (Original submission)

Review form: Reviewer 1

Is the manuscript scientifically sound in its present form?

Yes

Are the interpretations and conclusions justified by the results?

Yes

Is the language acceptable?

Yes

Is it clear how to access all supporting data?

Not Applicable

Do you have any ethical concerns with this paper?

No

Have you any concerns about statistical analyses in this paper?

No

Recommendation?

Accept with minor revision (please list in comments)

Comments to the Author(s)

This manuscript evaluated the variations of thermal properties and falling film heat transfer performance of nanoparticle-enhanced ionic liquids in an absorption refrigeration cycle. The results provide guidance to develop new working pairs for absorption refrigeration. It is recommended to be accepted after some English writing errors are corrected and some questions are answered.

1. It should be " To improve the thermal conduction " in Row 23 in Page 1. Please check it.
2. "The variation....." should be " The variations....." in Row 25 in Page 1.
3. "was then measured over a wide range....." should be " were then measured over wide ranges....." in Row 26 in Page 1.
4. "improved with increasing mass fraction" should be " increased with increasing mass fraction " in Row 29 in Page 1.
5. "refrigeration, which" should be " refrigeration that" in Row 36 in Page 1.
6. "efficiency, to" should be "efficiency to " in Row 109 in Page 6.
7. The grammar of the sentence "In an absorption refrigerator, the refrigerants from the evaporator are absorbed by and desorbed from the absorbents in the absorber and desorber, respectively, which are exothermic and endothermic processes." in Row 38-39 of page 1 may be incorrect.
8. Please provide the meshing tool and the version of the Fluent.
9. The reference No. in Figure should be [22] according to your statement.
10. Have you considered the mass transfer in your simulation? If 'no', it did not match the absorption condition.
11. Please provide the numerical model such as governing equations.
12. In Section 3.4, the liquid inlet is better to be a slot instead of a hole due to the present two dimensions domain.
13. Please explain the grate deviation between the measured data and the Batchelor model in Fig.8.
14. It is confused that the variation of local heat transfer coefficient with peripheral angle in Fig. 4 is fluent but fluctuated strongly in Fig.15.
15. The numerical simulations just were conducted for two conditions, which can be done for more cases.
16. A nomenclature is necessary for this study, in which all abbreviations should be covered.
17. "Variation" should be "Variations" in the title.
18. Why the effect of temperature was not reflected in each fitted correlation. After all, the variations of thermal properties with temperature have been concerned.

Review form: Reviewer 2 (Patrice Estellé)**Is the manuscript scientifically sound in its present form?**

No

Are the interpretations and conclusions justified by the results?

No

Is the language acceptable?

Yes

Is it clear how to access all supporting data?

No

Do you have any ethical concerns with this paper?

No

Have you any concerns about statistical analyses in this paper?

No

Recommendation?

Major revision is needed (please make suggestions in comments)

Comments to the Author(s)

The paper is suitable for the journal and reports an experimental investigation of thermophysical properties of ionanofluids. In addition, heat transfer performance in falling film flow on a horizontal tube is also performed.

At this stage, the paper requires amendments and clarifications before possible publication. All comments and questions to consider are here listed:

Precise if data of Table 1 come from the manufacturer or were evaluated by the authors.

For GNs, indicate if data come from the manufacturer or were evaluated by the authors. In the second case explain how they have been evaluated.

Indicate the type of device used for ultrasonication, brand, power, frequency, temperature.

Measurements methods have to be better described to make the work reproducible. In particular, include measured uncertainties with calibration fluids, for viscosity measurement give the type of geometry used, their dimensions, measurement has to be performed under steady-state conditions, without measurements are not reliable. Flow curves must be shown, not only the viscosity values to evidence the Newtonian behavior (or not).

eq2 is defined for spheres; GNs do not have this shape consequently this model is not scientifically relevant.

Eq5 does not give any physical insight, to remove, eq6 is sufficient and well predicted the data.

Where come from the density value of table 4?

To be relevant, the comparison of local heat coefficient in figure 15 must be performed considering uncertainty analysis of all experimental parameters.

minor comments: include error bars in figures

Decision letter (RSOS-182040.R0)

06-Feb-2019

Dear Dr Zhang:

Title: Variation of thermophysical properties and heat transfer performance of nanoparticle-enhanced

ionic liquids

Manuscript ID: RSOS-182040

The editor assigned to your manuscript has now received comments from reviewers. We would like you to revise your paper in accordance with the referee and Subject Editor suggestions which can be found below (not including confidential reports to the Editor). Please note this decision does not guarantee eventual acceptance.

Please submit your revised paper before 01-Mar-2019. Please note that the revision deadline will expire at 00.00am on this date. If we do not hear from you within this time then it will be assumed that the paper has been withdrawn. In exceptional circumstances, extensions may be possible if agreed with the Editorial Office in advance. We do not allow multiple rounds of revision so we urge you to make every effort to fully address all of the comments at this stage. If deemed necessary by the Editors, your manuscript will be sent back to one or more of the original reviewers for assessment. If the original reviewers are not available we may invite new reviewers.

Please also include the following statements alongside the other end statements. As we cannot publish your manuscript without these end statements included, if you feel that a given heading is not relevant to your paper, please nevertheless include the heading and explicitly state that it is not relevant to your work.

- Ethics statement

Please clarify whether you received ethical approval from a local ethics committee to carry out your study. If so please include details of this, including the name of the committee that gave consent in a Research Ethics section after your main text. Please also clarify whether you received informed consent for the participants to participate in the study and state this in your Research Ethics section.

OR

Please clarify whether you obtained the necessary licences and approvals from your institutional animal ethics committee before conducting your research. Please provide details of these licences and approvals in an Animal Ethics section after your main text.

OR

Please clarify whether you obtained the appropriate permissions and licences to conduct the fieldwork detailed in your study. Please provide details of these in your methods section.

- Acknowledgements

On behalf of the Subject Editor Professor Anthony Stace and the Associate Editor Professor Hazel Cox.

RSC Associate Editor:
Comments to the Author:
(There are no comments.)

RSC Subject Editor:
Comments to the Author:
(There are no comments.)

Reviewers' Comments to Author:
Reviewer: 1

Comments to the Author(s)

This manuscript evaluated the variations of thermal properties and falling film heat transfer performance of nanoparticle-enhanced ionic liquids in an absorption refrigeration cycle. The results provide guidance to develop new working pairs for absorption refrigeration. It is recommended to be accepted after some English writing errors are corrected and some questions are answered.

1. It should be " To improve the thermal conduction " in Row 23 in Page 1. Please check it.
2. "The variation....." should be " The variations....." in Row 25 in Page 1.
3. "was then measured over a wide range....." should be " were then measured over wide ranges....." in Row 26 in Page 1.
4. "improved with increasing mass fraction" should be " increased with increasing mass fraction " in Row 29 in Page 1.
5. "refrigeration, which" should be " refrigeration that" in Row 36 in Page 1.
6. "efficiency, to" should be "efficiency to " in Row 109 in Page 6.
7. The grammar of the sentence "In an absorption refrigerator, the refrigerants from the evaporator are absorbed by and desorbed from the absorbents in the absorber and desorber, respectively, which are exothermic and endothermic processes." in Row 38-39 of page 1 may be incorrect.
8. Please provide the meshing tool and the version of the Fluent.

9. The reference No. in Figure should be [22] according to your statement.
10. Have you considered the mass transfer in your simulation? If 'no', it did not match the absorption condition.
11. Please provide the numerical model such as governing equations.
12. In Section 3.4, the liquid inlet is better to be a slot instead of a hole due to the present two dimensions domain.
13. Please explain the grate deviation between the measured data and the Batchelor model in Fig.8.
14. It is confused that the variation of local heat transfer coefficient with peripheral angle in Fig. 4 is fluent but fluctuated strongly in Fig.15.
15. The numerical simulations just were conducted for two conditions, which can be done for more cases.
16. A nomenclature is necessary for this study, in which all abbreviations should be covered.
17. "Variation" should be "Variations" in the title.
18. Why the effect of temperature was not reflected in each fitted correlation. After all, the variations of thermal properties with temperature have been concerned.

Reviewer: 2

Comments to the Author(s)

The paper is suitable for the journal and reports an experimental investigation of thermophysical properties of ionanofluids. In addition, heat transfer performance in falling film flow on a horizontal tube is also performed.

At this stage, the paper requires amendments and clarifications before possible publication. All comments and questions to consider are here listed:

Precise if data of Table 1 come from the manufacturer or were evaluated by the authors.

For GNs, indicate if data come from the manufacturer or were evaluated by the authors. In the second case explain how they have been evaluated.

Indicate the type of device used for ultrasonication, brand, power, frequency, temperature.

Measurements methods have to be better described to make the work reproducible. In particular, include measured uncertainties with calibration fluids, for viscosity measurement give the type of geometry used, their dimensions, measurement has to be performed under steady-state conditions, without measurements are not reliable. Flow curves must be shown, not only the viscosity values to evidence the Newtonian behavior (or not).

eq2 is defined for spheres; GNs do not have this shape consequently this model is not scientifically relevant.

Eq5 does not give any physical insight, to remove, eq6 is sufficient and well predicted the data.

Where come from the density value of table 4?

To be relevant, the comparison of local heat coefficient in figure 15 must be performed considering uncertainty analysis of all experimental parameters.

minor comments: include error bars in figures

Author's Response to Decision Letter for (RSOS-182040.R0)

See Appendix A.

RSOS-182040.R1 (Revision)

Review form: Reviewer 1

Is the manuscript scientifically sound in its present form?

Yes

Are the interpretations and conclusions justified by the results?

Yes

Is the language acceptable?

Yes

Is it clear how to access all supporting data?

Not Applicable

Do you have any ethical concerns with this paper?

No

Have you any concerns about statistical analyses in this paper?

No

Recommendation?

Accept with minor revision (please list in comments)

Comments to the Author(s)

This manuscript has been well revised, and is recommended to be published.

Decision letter (RSOS-182040.R1)

19-Mar-2019

Dear Dr Zhang:

Title: Variations of thermophysical properties and heat transfer performance of nanoparticle-enhanced

ionic liquids

Manuscript ID: RSOS-182040.R1

It is a pleasure to accept your manuscript in its current form for publication in Royal Society Open Science. The chemistry content of Royal Society Open Science is published in collaboration with the Royal Society of Chemistry.

RSC Associate Editor:
Comments to the Author:
(There are no comments.)

RSC Subject Editor:
Comments to the Author:
(There are no comments.)

Reviewer(s)' Comments to Author:
Reviewer: 1

Comments to the Author(s)
This manuscript has been well revised, and is recommended to be published.

Appendix A

Response to Referees

Dear Editors and Reviewers:

Thank you very much for your great contribution and for the reviewers' comments concerning our manuscript entitled "Variations of thermophysical properties and heat transfer performance of nanoparticle-enhanced ionic liquids "(ID: RSOS-182040). Those comments are all valuable and very helpful for revising and improving our paper, as well as the important guiding significance to our researchers. We have studied all comments carefully and have made correction comprehensively in hope of meeting approval. Revised portions are marked in red in the paper. Furthermore, each comments of two reviewers is listed below and is followed by our response and revision one by one in blue.

Reviewer #1:

Q1: It should be " To improve the thermal conduction " in Row 23 in Page 1. Please check it.

A1: Thanks for your correction. We have modified this sentence according to the reviewer's suggestion.

Q2: "The variation....." should be " The variations....." in Row 25 in Page 1.

A2: Thanks for your correction. We have modified this sentence according to the reviewer's suggestion.

Q3: "was then measured over a wide range....." should be " were then measured over wide ranges....." in Row 26 in Page 1.

A3: Thanks for your correction. We have modified this sentence according to the reviewer's suggestion.

Q4: "improved with increasing mass fraction" should be " increased with increasing mass fraction " in Row 29 in Page 1.

A4: Thanks for your correction. We have modified this sentence according to the reviewer's suggestion.

Q5: "refrigeration, which" should be " refrigeration that" in Row 36 in Page 1.

A5: Thanks for your correction. We have modified this sentence according to the reviewer's suggestion.

Q6: "efficiency, to" should be "efficiency to " in Row 109 in Page 6.

A6: Thanks for your correction. We have modified this sentence according to the reviewer's suggestion.

Q7: The grammar of the sentence "In an absorption refrigerator, the refrigerants from the

evaporator are absorbed by and desorbed from the absorbents in the absorber and desorber, respectively, which are exothermic and endothermic processes.” in Row 38-39 of page 1 may be incorrect.

A7: Thanks for your valuable suggestion. We have re-written this sentence according to the reviewer's suggestion in the revised manuscript as follow:

In an absorption refrigerator, the absorber takes in the refrigerant from the evaporator and thereafter releases it to the condenser in the desorber accompanied by exothermic and endothermic effects.

Q8: Please provide the meshing tool and the version of the Fluent.

A8: Thanks for your valuable suggestion. As you suggested, the meshing tool and the version of the Fluent were added in the revised manuscript as follow:

Considering that the most frequently used flow mode in the absorber and desorber units is falling film flow on a horizontal tube and its symmetrical structure, the physical model of falling film flow on half of the horizontal tube based on Gambit is depicted in Fig. 3.

The volume of fluid (VOF) model was selected for the simulations, which were performed using the Fluent software (version 6.3.26).

Q9: The reference No. in Figure should be [22] according to your statement.

A9: Thanks for your correction. We are very sorry for our incorrect writing. As the reviewer #2 suggested, the sources of the data listed in Table 1 are added and indicated in the revised manuscript. The reference No. in figure 15 was modified to [25] according to the statement in the revised manuscript as follow:

Figure 4 shows a comparison of the local heat transfer coefficient of water between the present results and the reference results[25].

Figure 4. Comparison of the local heat transfer coefficient of water between the present and reference results.

The serial number of remaining references has been updated in the revised version.

Q10: *Have you considered the mass transfer in your simulation? If 'no', it did not match the absorption condition.*

A10: Thanks for your valuable suggestion. In fact, we did not consider the mass transfer in our simulation. As is said in the introduction, during a practical absorption refrigeration cycle, the processes of absorption by and desorption from the absorbent are often performed under cooling and heating, respectively. The cooling and heating efficiency directly affects the absorption and desorption efficiency. Therefore, it would be of great value to enhance the thermal conductivity of ionic liquid absorbents such as [EMIm]Ac to improve its transport properties that influence the heat transfer. This manuscript mainly concerned the variations of thermophysical properties and heat transfer performance of the nanoparticle-enhanced ionic liquids. The coupled heat and mass transfer is not conducive to the direct analysis of the variations of thermophysical properties on heat transfer performance, because of that the mass transfer process is accompanied by the changes in the physicochemical properties of liquid film and the absorption and release of heat. So the mass transfer is not considered in my simulation. The suggestion is meaningful. We will consider the improved thermal conductivity with nanoparticles on mass transfer performance in our future research.

Q11: *Please provide the numerical model such as governing equations.*

A11: Considering the reviewer's valuable suggestion, the governing equations of the numerical model was added in the revised manuscript as follow:

The governing equations can be expressed as follows[25]

$$\nabla(\mathbf{u}) = 0 \quad (1)$$

$$\frac{\partial(\rho\mathbf{u})}{\partial t} + \nabla(\rho\mathbf{u} \cdot \mathbf{u}) = \nabla(\mu\nabla\mathbf{u}) - \nabla(p) + \rho\mathbf{g} + \mathbf{F} \quad (2)$$

$$\frac{\partial(\rho T)}{\partial t} + \nabla(\rho\mathbf{u}T) = \nabla\left(\frac{\lambda}{c_p} \nabla T\right) \quad (3)$$

Where \mathbf{u} is the velocity vector, $\rho\mathbf{g}$ is the gravitational force, \mathbf{F} is the external body force of surface tension, T is temperature.

Q12: In Section 3.4, the liquid inlet is better to be a slot instead of a hole due to the present two dimensions domain.

A12: Thanks for your correction. We have modified this sentence according to your suggestion. The liquid inlet was modified to a slot.

Q13: Please explain the grate deviation between the measured data and the Batchelor model in Fig.8.

A13: Thanks for your valuable suggestion. As suggested by the reviewer #2 that, the Batchelor

model is defined for spheres; GNs have sheet structure. So the Batchelor model is not scientifically relevant. For graphene nanoplatelet, Sarsam^[1] studied the thermophysical properties of water-based nanofluids containing triethanolamine-treated graphene nanoplatelets (TEA-GNP). The measured values of viscosity at 30 and 40 °C for the TEA-GNP 750 nanofluids were compared with the classical models of Einstein, Brinkman, and Batchelor show in Fig.1. A larger deviation is also seen in Fig.1. Considering the relatively large deviation, He developed a correlation by modifying the factor in the original Einstein model for spheres from a value of 2.5 to a fitted value of 550 that represents the TEA-GNPs of this study.

Fig.1 Comparison between the measured values of viscosity for different weight concentrations of water-based TEA-GNP 750 nanofluids at (a) 30 °C and (b) 40 °C with the classical models for viscosity of Einstein, Brinkman, and Batchelor and with the suggested correlation.

Considering the reviewer's valuable suggestion, we added the comparison between the measured values of viscosity with the classical models of Einstein, Brinkman not only the Batchelor model. A suggested correlation was also developed. The part in the manuscript have been modified as follow:

Furthermore, the measured values of viscosity were compared with the classical models of Einstein, Brinkman, Batchelor [27] and a suggested correlation was developed by modifying the factor in the original Einstein model for spheres from a value of 2.5 to a fitted value of 1.1 shown in Fig.8, which can well represents the GNPs in our study except at a lower mass fraction (with maximum deviation of 9.5%) due to a dominant self-lubrication effect of GNPs at lower mass fractions[26].

$$\text{Einstein model } \frac{\eta_{INF}}{\eta_{BL}} = 1 + 2.5\varphi \quad (5)$$

$$\text{Brinkman model } \frac{\eta_{INF}}{\eta_{BL}} = \frac{1}{(1-\varphi)^{2.5}} \quad (6)$$

$$\text{Batchelor model } \frac{\eta_{INF}}{\eta_{BL}} = 6.5\varphi^2 + 2.5\varphi + 1 \quad (7)$$

$$\text{Suggested correlation } \frac{\eta_{INF}}{\eta_{BL}} = 1 + 1.1\varphi \quad (8)$$

$$\varphi = \frac{\omega \rho_{\text{INF}}}{\rho_{\text{NP}}} \quad (9)$$

Where η_{INF} is the thermal conductivity of the INF, η_{BL} is the thermal conductivity of the BL, φ is the particle volume fraction calculated using Eq. (9), ω is the mass fraction of the INF, and ρ_{NP} and ρ_{INF} are the densities of the nanoparticle and the INF, respectively.

Figure 8. Comparison between the measured values of viscosity for different volume fraction of INF at 303.15 K with the classical models for viscosity of Einstein, Brinkman, and Batchelor and with the suggested correlation.

1. Sarsam WS, Amiri A, Zubir MNM, Yarmand H, Kazi SN, Badarudin A. 2016 Stability and thermophysical properties of water-based nanofluids containing triethanolamine-treated graphene nanoplatelets with different specific surface areas. *Colloids and Surfaces A: Physicochem. Eng. Aspects* **500**, 17–31.

Q14: It is confused that the variation of local heat transfer coefficient with peripheral angle in Fig. 4 is fluent but fluctuated strongly in Fig.15.

A14: Thanks for your valuable suggestion. In fact, Fig.4 was used to verify the reliability of the model in this paper with the reference [25] in the manuscript. The working media and boundary conditions of the liquid film flow rate on one side of the tube per unit length Γ and thermal condition for the cases in Fig. 4 and Fig.15 are different. Due to the unique physical properties of larger viscosity for ionic liquids, the liquid film flow rate Γ for the cases in Fig.15 was $0.3 \text{ kg}\cdot\text{m}^{-1}\cdot\text{s}^{-1}$ which is larger than that of water in Fig.4. ($\Gamma=0.168 \text{ kg}\cdot\text{m}^{-1}\cdot\text{s}^{-1}$); and the thermal condition is constant heat flux and constant wall temperature, respectively, for the cases in Fig. 4 and Fig.15. The fluctuation is also seen for certain cases in reference [25] shown in Fig.1.

Fig.1 Local heat transfer coefficient versus peripheral angle under different tube diameters.

In the manuscript, the local heat transfer coefficient is calculated when the computation reaches stability. Considering the reviewer's valuable suggestion, the local heat transfer coefficient is time-averaged according to reference [25] to reduce the fluctuation, and the result is shown in Fig.15. The different boundary conditions were added and indicated in the Fig.4 and Fig.15. The description for Fig.15 in the manuscript is modified as follow:

Figure 15 shows a comparison of the time-averaged local heat transfer coefficients of the falling films using the pure IL and the INF with mass fractions of 0.05%, 0.3%, 2%, 5% as the absorbent.

Figure 15. Local heat transfer coefficients of the falling film with different mass fractions of GNP.

*Q15:*The numerical simulations just were conducted for two conditions, which can be done for more cases.

A15: Thanks for your valuable suggestion. As you suggested, the ionanofluids with GNP mass fraction of 0%, 0.05%, 0.3%, 2%, 5% were done and is shown in Fig.15.

Figure 15. Local heat transfer coefficients of the falling film with different mass fractions of GNP.

Q16: A nomenclature is necessary for this study, in which all abbreviations should be covered.

A16: Thanks for your valuable suggestion. A nomenclature is added and covered all the abbreviations in the revised manuscript as follow:

Nomenclature

C_p specific heat capacity, $\text{J}\cdot\text{kg}^{-1}\cdot\text{K}^{-1}$

D tube diameter, m

F body force, N

h heat transfer coefficient, $\text{W}\cdot\text{m}^{-2}\cdot\text{K}^{-1}$

g gravity acceleration, $\text{m}\cdot\text{s}^{-2}$

q heat flux density, $\text{W}\cdot\text{m}^{-2}$

T temperature, K

\mathbf{u} velocity vector, $\text{m}\cdot\text{s}^{-1}$

Greek

λ thermal conductivity, $\text{W}\cdot\text{m}^{-1}\cdot\text{K}^{-1}$

μ dynamic viscosity, $\text{kg}\cdot\text{m}^{-1}\cdot\text{s}^{-1}$

ρ density, $\text{kg}\cdot\text{m}^{-3}$

ϕ volume fraction

Γ liquid film flow rate on one side of the tube per unit length, $\text{kg}\cdot\text{m}^{-1}\cdot\text{s}^{-1}$

Abbreviations

BL base liquid

GNP graphene nanoplatelet

IL ionic liquid

Ionanofluid ionic-liquid-based nanofluid

INF ionanofluid

Q17: “Variation” should be “Variations” in the title.

A17: Thanks for your correction. We are very sorry for our incorrect writing. We have modified this sentence according to your suggestion.

Q18: Why the effect of temperature was not reflected in each fitted correlation. After all, the variations of thermal properties with temperature have been concerned.

A18: Thanks for your valuable suggestion. Given the small change of thermal conductivity with increasing temperature, we provide the fitted correlation for viscosity and specific heat capacity neglecting that for the thermal conductivity. Considering the reviewer's suggestion, the fitted correlation between the thermal conductivity and temperature is added in the revised manuscript as follow:

In addition, a linear equation (Eq.(10)) was used to fit the experimentally measured thermal conductivity data. Table 3 summarized the values of the fitting parameters B₀ and B₁ for the various mass fractions.

$$\lambda = B_0 + B_1T \quad (10)$$

Table 3. Fitting parameters for the thermal conductivity data.

Mass fraction	Parameter		R ²
	B ₀	B ₁	
0% (BL)	0.2113	2.75146E-5	0.9814
0.05%	0.22705	4.72167E-5	0.99584
0.1%	0.23324	5.78453E-5	0.98808
0.3%	0.222	1.30405E-4	0.98572
0.5%	0.22774	1.17001E-4	0.99912
1%	0.23819	1.05179E-4	0.99457
2%	0.266	7.25622E-5	0.99677
3%	0.27571	6.60535E-5	0.9871
4%	0.27836	8.00711E-5	0.99661
5%	0.28515	8.69292E-5	0.99291

The serial number of remaining equations and tables has been updated in the revised version.

Special thanks to you for your good comments.

Reviewer #2:

Q1: Precise if data of Table 1 come from the manufacturer or were evaluated by the authors.

A1: Thanks for your valuable suggestion. The data of Table 1 were evaluated in this work and the measurement methods were illustrated in Sec.3.3. As you suggested, the data sources were

indicated in Table 1 and some description in the manuscript is modified as follow:

[EMIm]Ac (purity \geq 98%, water content \leq 1%) was purchased from the Lanzhou Institute of Chemical Physics, Chinese Academy of Sciences. The structural formula of [EMIm]Ac is depicted in Fig. 1 and its thermophysical properties measured in this work at the standard temperature of 293.15 K are summarized in Table 1. The minor deviations in certain physical properties were ascribed to the different manufacturing processes used by different suppliers.

Table1. Physical properties of [EMIm]Ac at 293.15 K.

Property	Present study	Literature	Deviation
Molecular weight	170.2	–	–
Density ($\text{g}\cdot\text{cm}^{-3}$)	1.10493	1.10302[20]	0.17%
Viscosity ($\text{mPa}\cdot\text{s}$)	155.1	162[21]	4.26%
Specific heat capacity ($\text{J}\cdot\text{kg}^{-1}\cdot\text{K}^{-1}$)	1868	1625[12]	14.5%
Thermal conductivity($\text{W}\cdot\text{m}^{-1}\cdot\text{K}^{-1}$)	0.221	0.211[22]	4.7%

Q2: For GNs, indicate if data come from the manufacturer or were evaluated by the authors. In the second case explain how they have been evaluated.

A2: Thanks for your valuable suggestion. The data for GNs come from the manufacturer and we have indicated the data sources in the manuscript as follow:

The GNPs exhibited a thermal conductivity of $3,000 \text{ W}\cdot\text{m}^{-1}\cdot\text{K}^{-1}$, a diameter of 5–10 μm , a thickness of 4–20 nm and a density of $0.6 \text{ g}\cdot\text{cm}^{-3}$ and consisted of less than 20 layers (the data come from the manufacturer).

Q3: Indicate the type of device used for ultrasonication, brand, power, frequency, temperature.

A3: Thanks for your valuable suggestion. Considering the Reviewer's suggestion, the type of device used for ultrasonication, brand, power, frequency, temperature were added in the revised manuscript as follows:

Various mass fractions of the GNPs were dispersed in [EMIm]Ac using a constant temperature magnetic stirrer for 60 min followed ultrasonication for 60 min at 25°C by ultrasonic cleaning machine (C15, XIERBAO, Beijing, power: 300W, frequency: 40KHz) for 60 min, affording INFs with mass fractions of 0.05%, 0.1%, 0.3%, 0.5%, 1%, 2%, 3%, 4% and 5%.

Q4: Measurements methods have to be better described to make the work reproducible. In particular, include measured uncertainties with calibration fluids, for viscosity measurement give the type of geometry used, their dimensions, measurement has to be performed under steady-state conditions, without measurements are not reliable. Flow curves must be shown, not only the viscosity values to evidence the Newtonian behavior (or not).

A4: Thanks for your valuable suggestion. In this paper, the shear rate of 500 $1/\text{s}$ was selected to avoid the Taylor vortices area for steady-state shear testing. As you suggested, the measured uncertainties with calibration fluids, the type of geometry used and their dimensions for viscosity measurement, a figure about the flow curves and some description were added in the

revised manuscript as follows:

Thermal conductivities were evaluated using a laser thermal conductivity meter (LFA 467, NETZSCH, Germany) by the flash method over the temperature range of 293.15–373.15K. Pyroceram 9606 provided by the supplier was used to calibrate the meter with a relative uncertainty of 3%. Viscosities were measured using a viscotester (Viscotester iQ, HAAKE, Germany) over the temperature range of 283.15–373.15 K. The viscotester was calibrated by the standard viscosity liquid provided by the supplier with a relative uncertainty of 0.65%. The torque resolution is 0.01 mN·m. The type of geometry used is cylinder double-gap (inner cylinder: inner diameter and outer diameter is 20.810 mm and 21.281 mm; outer cylinder: inner diameter and outer diameter is 26.594 mm and 27.200 mm; height: 40mm; distance: 4mm; volume: 3cm³). The shear rate of 500 1/s was selected to avoid the Taylor vortices area for steady-state shear testing. Specific heat capacities were determined using a differential scanning calorimeter (DSC214Polyma, Co. NETZSCH, Germany) based on the sapphire method over the temperature range of 303.15–383.15 K. This was calibrated using Sapphire provided by the supplier with a relative uncertainty of 0.5%. Thermal stabilities were analyzed using the same differential scanning calorimeter as above, and the samples were heated from –50 °C to 350 °C at a rate of 10 °C/min under a nitrogen atmosphere. Densities were measured using a densimeter (DMA5000M-Lovis200M, Anton Paar Co.Austria). This was calibrated using air and ultra pure water provided by Anton Paar GmbH and compared with values reported in the densimeter instruction manual. The found uncertainty was less than $\pm 1 \cdot 10^{-5}$ g·cm⁻³ and the accuracy is 5×10^{-6} g·cm⁻³.

The shear stress are plotted in Fig.6 (a) as a function of shear rate for the samples within the shear rate range of 1 1/s–500 1/s at 293.15 K. From which, it can be found that the behavior of ionanofluids were quite Newtonian when the mass fraction of GNPs is lower than 0.5%. However, the behavior of ionanofluids were quite non-Newtonian when the mass fraction of GNPs is larger than 0.5%. The variation of the natural logarithm of the viscosity (tested with a shear rate of 500 1/s) of the BL and the INFs with mass fractions of 0.05%,0.3%, 0.5%, 1%, 2%, 3% and 5% as a function of temperature is shown in Fig. 6 (b).

Figure 6. (a) The flow curves of the viscosity for the INFs,

The serial number of remaining figures has been updated in the revised version.

Q5: *eq2 is defined for spheres; GNs do not have this shape consequently this model is not scientifically relevant.*

A5: Thanks for your valuable suggestion. It is really true as reviewer suggested that, the Batchelor model is defined for spheres; for graphene nanoplatelet, Sarsam^[1] studied the thermophysical properties of water-based nanofluids containing triethanolamine-treated graphene nanoplatelets (TEA-GNP). The measured values of viscosity at 30 and 40 °C for the TEA-GNP 750 nanofluids were compared with the classical models of Einstein, Brinkman, and Batchelor show in Fig.1. Considering the relatively large deviation, he developed a correlation by modifying the factor in the original Einstein model for spheres from a value of 2.5 to a fitted value of 550 that represents the TEA-GNPs of this study.

Fig.1 Comparison between the measured values of viscosity for different weight concentrations of water-based TEA-GNP 750 nanofluids at (a) 30 °C and (b) 40 °C with the classical models for viscosity of Einstein, Brinkman, and Batchelor and with the suggested correlation.

Considering the reviewer's valuable suggestion, we added the comparison between the measured values of viscosity with the classical models of Einstein, Brinkman, not only the Batchelor model. A suggested correlation was also developed. The part in the manuscript have

been modified as follow:

Furthermore, the measured values of viscosity were compared with the classical models of Einstein, Brinkman, Batchelor [27] and a suggested correlation was developed by modifying the factor in the original Einstein model for spheres from a value of 2.5 to a fitted value of 1.1 shown in Fig.8, which can well represents the GNPs in our study except at a lower mass fraction (with maximum deviation of 9.5%) due to a dominant self-lubrication effect of GNPs at lower mass fractions[26].

$$\text{Einstein model } \frac{\eta_{INF}}{\eta_{BL}} = 1 + 2.5\varphi \quad (5)$$

$$\text{Brinkman model } \frac{\eta_{INF}}{\eta_{BL}} = \frac{1}{(1-\varphi)^{2.5}} \quad (6)$$

$$\text{Batchelor model } \frac{\eta_{INF}}{\eta_{BL}} = 6.5\varphi^2 + 2.5\varphi + 1 \quad (7)$$

$$\text{Suggested correlation } \frac{\eta_{INF}}{\eta_{BL}} = 1 + 1.1\varphi \quad (8)$$

$$\varphi = \frac{\omega \rho_{INF}}{\rho_{NP}} \quad (9)$$

Where η_{INF} is the thermal conductivity of the INF, η_{BL} is the thermal conductivity of the BL, φ is the particle volume fraction calculated using Eq. (9), ω is the mass fraction of the INF, and ρ_{NP} and ρ_{INF} are the densities of the nanoparticle and the INF, respectively.

Figure 8. Comparison between the measured values of viscosity for different volume fraction of INF at 303.15 K with the classical models for viscosity of Einstein, Brinkman, and Batchelor and with the suggested correlation.

1. Sarsam WS, Amiri A, Zubir MNM, Yarmand H, Kazi SN, Badarudin A. 2016 Stability and thermophysical properties of water-based nanofluids containing triethanolamine-treated graphene nanoplatelets with different specific surface areas. *Colloids and Surfaces A:*

Physicochem. Eng. Aspects **500**, 17–31.

Q6: *Eq5 does not give any physical insight, to remove, eq6 is sufficient and well predicted the data.*

A6: Thanks for your valuable suggestion. Maybe Eq.5 does not give a novel physical insight, but it had been hoped to quantitatively bridge the experimental measured specific heat capacity for the [EMIm]Ac-based ionic liquids (with addition of GNP) and the temperature. Eq.6 is classic theoretical model for ionic liquids concerning the volume fraction of nanoparticle while neglecting the temperature. So the Eq.5 is retained in our opinion.

Q7: *Where come from the density value of table 4?*

A7: Thanks for your valuable suggestion. The density is evaluated in this paper using a densimeter (DMA5000M-Lovis200M, Anton Paar Co.Austria). The density measurement method is added in the Sec. 3.3 in the revised manuscript as follow:

Densities were measured using a densimeter (DMA5000M-Lovis200M, Anton Paar Co.Austria). This was calibrated using air and ultra pure water provided by Anton Paar GmbH and compared with values reported in the densimeter instruction manual. The found uncertainty was less than $\pm 1 \cdot 10^5 \text{ g} \cdot \text{cm}^{-3}$ and the accuracy is $5 \times 10^6 \text{ g} \cdot \text{cm}^{-3}$.

The number of the table is modified to table 5 because of the addition of the table of "Fitting parameters for the thermal conductivity data". The description for the table is modified in the revised manuscript as follow:

The properties of the working medium measured in this work are summarized in Table 5.

Q8: *To be relevant, the comparison of local heat coefficient in figure 15 must be performed considering uncertainty analysis of all experimental parameters.*

A8: Thanks for your valuable suggestion. In fact, the local heat coefficient for the ionic liquids in figure 15 is by numerical calculation. So the uncertainty analysis of all experimental parameters is not considered. The reliability of the numerical model is verified in Sec.3.4. Considering the reviewer's valuable suggestion, the uncertainty analysis of all experimental parameters were added in the Sec. 3.3 and the error bars in the figures were also added.

Q9: *minor comments: include error bars in figures*

A9: Thanks for your valuable suggestion. As you suggested, the measured error bars were added; and due to the wide range of y-coordinate for the viscosity and thermal conductivity, the detail view of the error bar is also given in the figure 6 (c) and figure 9. The measured error bars were added in the revised manuscript as follow:

Figure 6(c). Viscosity of the INFs versus mass fraction.

Figure 9. Thermal conductivity of the INFs versus temperature.

Figure 12. Specific heat capacity of the INFs as a function of temperature

Special thanks to you for your good comments.